# The association between HIV-related stigma, HIV knowledge and HIV late presenters among people living with HIV (PLHIV) attending public primary care clinic settings in Selangor

**Mohd Zulfikry Bin Ahmad**[1⊙], **Mazapuspavina Md. Yasin**[1]*, **Nafiza Mat Nasir**[1⊙], **Mariam Mohamad**[2⊙]

1 Department of Primary Care Medicine, Faculty of Medicine, Universiti Teknologi MARA, Selangor, Malaysia, 2 Department of Population Health and Preventive Medicine, Faculty of Medicine, Universiti Teknologi MARA, Selangor, Malaysia

⊙ These authors contributed equally to this work.

* puspavina@uitm.edu.my

**Data Availability Statement:** All relevant data are within the paper and its Supporting information files.

## Abstract

### Introduction

HIV late presenters were defined as individuals presenting with a CD4 count below 350 cells/µL or with an AIDS-defining event, according to the European Late Presenter Consensus working group. Early diagnosis and treatment of HIV have proven beneficial for people living with HIV (PLHIV), reducing the burden on healthcare systems, and contributing to ending the HIV/AIDS epidemic. However, in Malaysia, over 50% of newly diagnosed HIV patients present late, leading to increased morbidity and premature mortality. This study aims to determine the prevalence of late HIV presenters and its association with HIV-related stigma and HIV knowledge among PLHIV attending public primary care clinics in Selangor.

### Methods

A cross-sectional study was conducted at selected public health clinics in Selangor, involving PLHIV aged 18 years and older, who were diagnosed since 2019. HIV-related stigma was measured using the Malay version of Berger's HIV Stigma Scale, and HIV knowledge was assessed using the Malay version of Brief HIV-KQ-18. Univariate and multivariate logistic regression analyses were performed to identify factors associated with late HIV presentation.

### Results

A total of 400 participants were included in the study, with 60.0% (n = 240, 95% CI: 55.0–65.0) classified as late presenters. The participants had a mean age of 30.29 (±7.77) years. The risk factors for late presenters were high levels of HIV-related stigma (aOR = 1.049, 95% CI: 1.034–1.063, p-value <0.001), low levels of HIV knowledge (aOR = 0.709, 95% CI:

**Funding:** Funded studies Authors received award: Mazapuspavina Md Yasin, Nafiza Mat Nasir, Mariam Mohamad Grant number: 600-UiTMSEL (PI. 5/4) (024/2022) Geran Penyelidikan Dana UiTM Cawangan Selangor (DUCS) Malaysia URL: https://orchid.uitm.edu.my/irmis/index.php?val= loggedout The funders had no role in study design, data collection and analysis, decision to publish, or preparation of the manuscript.

**Competing interests:** The authors have declared that no competing interests exist.

0.646–0.778, p-value <0.001), tertiary education background (aOR = 15.962, 95% CI: 1.898–134.235, p-value = 0.011), and being single (aOR = 3.582, 95% CI: 1.393–9.208, p-value = 0.008).

## Conclusion

This study highlights the association between high levels of HIV-related stigma, low levels of HIV knowledge, and late HIV presentation. Interventions targeting stigma reduction and HIV education can promote early testing and prompt access to care, improving health outcomes for PLHIV.

## Introduction

In 2022, the estimated global population of people living with HIV (PLHIV) was approximately 39 million (33.1–45.7 million). The number of new infections stood at 1.3 million (1.0–1.7 million) people, indicating a decline from 2.1 million (1.6–2.8 million) people in 2010 [1]. A similar epidemiological pattern can be observed in Malaysia, where the incidence of new HIV infections has followed a discernible trend. Since the detection of the first case of AIDS in 1987 [2], the number of new infections has exhibited an upward trajectory, peaking at approximately 6,978 cases in 2002. However, following the introduction of the Harm Reduction Program, which includes initiatives such as the Needle and Syringe Exchange Program (NSEP) and Opioid Substitution Therapy (OST), there has been a rapid decline in new HIV infections. Subsequently, the number of new infections decreased to 3,177 cases in 2022 [3].

Despite a gradual decline in the number of new HIV infections, there remains a significant prevalence of late HIV diagnosis among affected individuals. In Malaysia, it is estimated that approximately 68% of new patients were diagnosed at a late stage in 2021 [4]. On a global scale, the prevalence of HIV late presenters ranges from 40% to 60% in developed countries and tends to be higher in developing countries, ranging from 60% to 86% [5, 6]. HIV late presenters can be identified based on either immunological criteria, specifically the CD4 count, or clinical criteria. The European Late Presenter Consensus working group has established a definition of late presenters as persons presenting for care with a CD4 count below 350 cells/µL or those who present with an AIDS-defining event, regardless of their CD4 count [7]. Similarly, in Malaysia, according to the Country Progress Report on Global AIDS Monitoring 2022, HIV late presenters were defined as individuals with a CD4 count less than 350 cells/mm3 [4].

Late HIV diagnosis has significant adverse outcomes for individuals, including physical and psychological impacts, while also increasing the risk of onward transmission in society [6, 8]. In Malaysia, the financial burden of HIV-related expenses falls predominantly on domestic public funding, covering more than 80% of total expenditure in 2021, approximately RM82.9 million [4]. Multiple studies have been conducted to understand the risk factors for HIV late presenters. These risk factors include older age, specifically among the aging population, being male, transmission through heterosexual contact, and involvement in injecting drug use [9, 10]. Furthermore, barriers to HIV testing were categorized into three main types; personal barriers, encompassing a lack of understanding about HIV/AIDS, along with the hesitancy to undergo testing due to perceived low risk of HIV and fear of the test outcome; healthcare service-related barriers, which involved a lack of confidence in healthcare providers; and social barriers, which included issues related to stigma, discrimination, and the lack of social support [11].

Stigma towards PLHIV involves prejudice, negative attitudes, and abuse. It leads to shame and discredits individuals in the eyes of others [12]. Stigma can be viewed from two perspectives, first is stigma experienced by PLHIV and second is stigma exhibited by the general population towards PLHIV. Our study is focusing on stigma perceived by PLHIV which can be categorized into enacted, anticipated and internalized stigma. Anticipated stigma was encountered by PLHIV due to prevailing HIV-related discrimination and stigma experienced by other PLHIV. Internalized stigma was influenced by individual moral judgements that linked HIV status with morally questionable behaviors, as well as their own negative self-assessment [13]. This stigma is multifaceted and often reinforced by associated HIV with marginalized behaviors like sex work, substance use and homosexual practices. In Malaysia, injecting drug users are labeled as criminals, those involved in prostitution are seen as immoral and sinful, and individuals with multiple sexual partners or homosexuality are branded as sexually immoral [14].

In Asia, with its rich cultural and religious values, PLHIV from high-risk behaviors face stigmatization and discrimination. These perspectives were observed within family, communities and healthcare settings. Examples of stigma and discrimination were negative labelling, separation of personal belongings, isolation of PLHIV, substandard care and rejection by healthcare providers, family and community members. The identified factors for stigma and discrimination were poor knowledge about HIV, fear of contracting HIV, own personal values and beliefs, religious thoughts, and sociocultural norms [15]. This indicate the need for HIV education across individual, family, community members, and healthcare providers to improve attitude, self-efficacy and acceptance of PLHIV [16].

Numerous studies in Malaysia have assessed HIV-related knowledge among the general population, indicating moderate to high level of awareness. The correct response rates for HIV knowledge ranged from 62.8% to 86.5% in different studies [17, 18]. It is important to note that the varying response rates could be attributed to differences in the sample demographics and the tools used to measure HIV knowledge. Saddki et al. found that the mean percentage of correct responses on HIV knowledge among PLHIV was 53.7% [19]. However, there remain limited studies specifically focusing on late presenters among PLHIV in Malaysia.

Furthermore, HIV related stigma and limited HIV knowledge have been identified as barriers to early testing, treatment, and prevention efforts [20, 21]. This pervasive stigma is a pressing concern globally, including in Malaysia. Nevertheless, our understanding of the extent of HIV related stigma and the level of HIV knowledge among PLHIV who are diagnosed late in Malaysia remains limited. Selangor, one of the major states in Malaysia, has the highest number of PLHIV, accounting for 30% of the total population [4]. Located in the densely populated Klang Valley region, it serves as a critical area for research. This research aims to determine the prevalence of HIV late presenters, and its association with HIV-related stigma, knowledge, and sociodemographic factors, among PLHIV who attend public primary care clinics in Selangor.

This study holds immense significance in addressing critical public health issues. By investigating the relationship between HIV-related stigma, knowledge, and late presentation for care, it sheds light on the intricate interplay of psychosocial factors that influence health-seeking behavior among PLHIV. The findings from this study could offer valuable insights into the reasons behind delayed diagnosis and treatment initiation, enabling the development of targeted interventions to reduce HIV-related stigma and enhance HIV knowledge, thereby encouraging early detection and timely access to healthcare services. Ultimately, such interventions have the potential to improve the overall outcomes of PLHIV, contributing significantly to the global efforts to end the AIDS epidemic.

## Materials and methods

### Study design

This was a cross-sectional study conducted among PLHIV who were attending HIV/STI clinics in selected health clinics in the central, urban region of Gombak and Petaling District, Selangor. The study utilized an anonymous, self-administered questionnaire and medical records for data collection. Gombak and Petaling Districts were chosen due to their high population density and substantial number of active PLHIV attending HIV/STI clinics in Selangor.

### Sample size determination

Multiple sample sizes were calculated based on different prevalence corresponding to the objectives. The determination of the sample size was based on the prevalence of HIV late presenters, HIV related stigma and HIV knowledge. The highest calculated sample size was selected for this study. The calculation was performed using the "Sample size for Proportion or Descriptive Study" tool from OpenEpi Version 3.01, available at https://www.openepi.com/SampleSize/SSPropor.htm. The calculation incorporated data from the Country Progress Report on HIV/AIDS 2019, provided by the Ministry of Health, Malaysia. According to the report, it was estimated that approximately 53% of PLHIV had an initial CD4 cell count less than 350 cells/mm$^3$ in 2018 [22]. With a significance level of 0.05, the estimated sample size required for the study was determined to be 383.

### Participant recruitment, sampling method, and data collection

The study included 400 patients diagnosed with HIV aged 18 years and older. The sample was selected using a convenience sampling method, based on the patients who attended the health facility for their appointment during the data collection period. To meet the inclusion criteria, participants needed to be PLHIV aged 18 years and older, irrespective of nationality, diagnosed with HIV from 2019 onward, have a baseline CD4 count at diagnosis, and possess the ability to read and understand the Malay language. Participants with mental instability or underlying any psychiatric illnesses that could impair their ability to answer the questionnaire were excluded. The recruitment of participants was carried out for a total of 10 months, from 1[st] July 2022 until 30[th] April 2023.

The tools used in this study included the Malay version of Berger's HIV Stigma Scale and the Malay version of Brief HIV Knowledge Questionnaire (HIV-KQ-18). Permission to use these questionnaires was obtained from their developers (S1 and S2 Figs). Both questionnaires had been validated and translated to the Malay language [19, 23–25]. The questionnaire was divided into four different sections: Section 1 captured sociodemographic profiles, Section 2 collected clinical characteristics, Section 3 assessed Berger's HIV Stigma Scale, and Section 4 assessed HIV-KQ-18.

Section 1 consisted of a self-administered questionnaire on sociodemographic profiles. It included six items, namely age, gender, ethnicity, nationality, marital status, and educational background.

Section 2 focused on clinical characteristics, including the mode of HIV transmission such as heterosexual, men who have sex with men (MSM), people who inject drugs (PWID), blood transfusion and unknown. Baseline CD4 count was collected in this section, and for patients who were unaware of their baseline CD4 count, it was obtained from their medical records.

Section 3 involved the Malay version of Berger's HIV Stigma Scale, which consists of 40 items organized into 4 subscales: personalized stigma (enacted stigma), disclosure concern (anticipated stigma), concerns about public attitudes (anticipated stigma) and negative self-

image (internalized stigma). The scale was rated on a 4-point Likert scale (strongly disagree, disagree, agree, strongly agree). Two items, specifically items 8 and 21, were reverse scored. Scores were calculated by adding up the raw values of the items within each subscale. Sixteen items were allocated to more than one subscale, reflecting the intercorrelations among the factors upon which the subscales are based. The possible score range depends on the number of items in the scale. For the total HIV Stigma Scale, scores can range from 40 to 160 (1x40 items to 4x40 items), with higher scores indicating a greater level of HIV-related stigma. The personalized stigma subscale can yield scores ranging from 18 to 72. The disclosure subscale has a score range from 10 to 40. The negative self-image subscale ranges from 13 to 52, while the public attitudes subscale ranges from 20 to 80. The Malay version demonstrated an internal consistency reliability of 0.92, which is comparable to the original English version's reliability of 0.96 [24].

Section 4 focused on the Malay version of the HIV-KQ-18 questionnaire. This questionnaire was developed to assess knowledge regarding HIV transmission, particularly in relation to sexual and casual contact [25]. It consisted of 18 items and was designed for low literacy populations [26]. The questionnaire was a simplified version of the HIV Knowledge Questionnaire (HIV-KQ), which contained 45 items. The HIV-KQ-18 was selected due to its good internal consistency, with Cronbach's α ranging from 0.78–0.89. It also demonstrates a strong association with the original and much longer items, HIV-KQ. Additionally, a Malay version of the questionnaire is available and has been validated, showing good internal consistency with a Cronbach's α value of 0.78 [19]. For each knowledge item, participants were provided with three response options: "true", "false", and "don't know". Correct responses were given 1 point, while incorrect or "don't know" responses were given 0 points. The total score ranged from 0 to 18, with higher scores indicating better knowledge.

Participants were given a questionnaire during the clinic visit. Clear written and verbal instructions were provided on how to complete the questionnaires. Participants were encouraged to seek clarification from the investigators at any time if any queries arose. They were also reminded to answer the questionnaires truthfully by themselves. On average, participants took approximately 15–20 minutes to complete the questionnaires. Once they finished, they handed the questionnaire to the investigator, who then checked for completeness. Fig 1 illustrates the study's conduct.

## Definitions of key variables

HIV late presenters were defined as individuals presenting for care with a CD4 count below 350 cells/μL or those who presented with an AIDS-defining event, regardless of their CD4 count. The CD4 count was determined using the baseline measurement within 6 months from the date of diagnosis. The gender, ethnicity, and nationality of the participants were recorded as stated on their identification cards. The education level was assessed based on the highest level of education attained by the participants. The income groups were categorized as low, middle, and high based on the income thresholds published by the Department of Statistics Malaysia (DOSM). The low-income group, known as B40, had an income threshold of less than RM4,849, the middle-income group, known as M40, had an income threshold between RM4,850 and RM10,959, and the high-income group, known as T20, had an income threshold of more than RM10,960 [27].

## Data analysis

The IBM Statistical Package for Social Sciences (SPSS) program version 29.0 was utilized for data entry and statistical analysis. We used the techniques developed by our colleagues [19] to

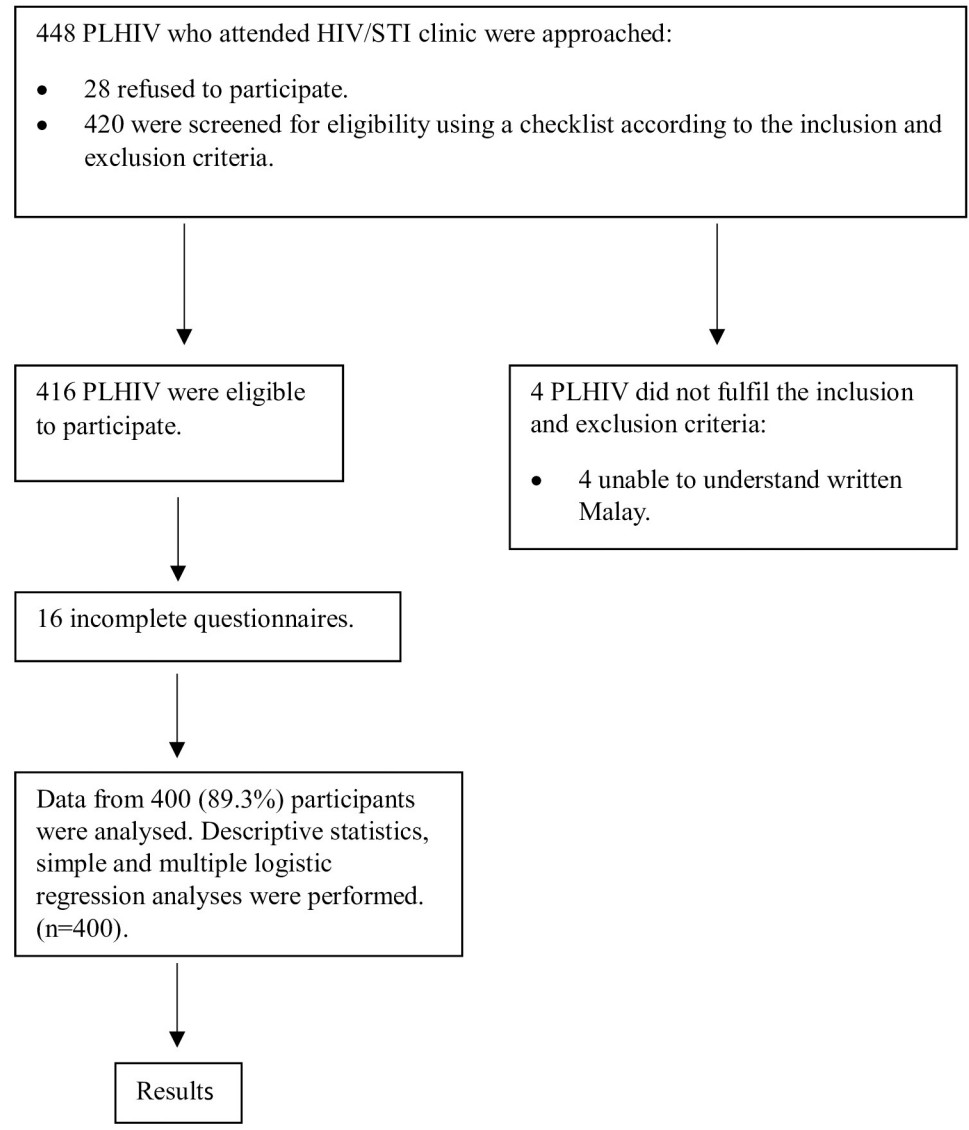

**Fig 1. Flow chart of the study's conduct.**

analyze the data. Descriptive statistics were used to describe the sociodemographic profile, clinical characteristics, level of HIV knowledge, and level of HIV-related stigma. Continuous data with a normal distribution were represented using the mean [+/- standard deviation (SD)], while non-normally distributed data were described using the median and interquartile ranges (IQR). Frequencies and percentages were used to describe categorical data.

Independent t-tests were used to compare the mean scores of the level of HIV-related stigma and the level of HIV knowledge between two groups: HIV late presenters and HIV non-late presenters. The level of statistical significance was set at a p-value of <0.05 with a 95% confidence interval.

The association between HIV late presenters with independent variables such as age, gender, ethnicity, nationality, marital status, education level, occupation, monthly income, mode of transmission, level of HIV-related stigma and level of HIV knowledge among PLHIV were

firstly analysed using simple logistic regression to screen important independent variables. Factors with a p-value <0.25 and clinically important variables such as age, gender, ethnicity, nationality, occupation, and monthly income, and mode of transmission were then selected to be included in the multiple logistic regression analysis (MLogR). MLogR was performed to control any potential confounding factors so that only the significant independent factors were identified to be associated with HIV late presentation (final model). Statistical significance was considered at P value <0.05.

The final model was checked for multicollinearity using Variance Inflation Factor (VIF). VIF less than 10 indicates no collinearity between the variables [28]. Interaction was checked by pairing each of the significant independent variables in the final model. If there is interaction, it will be included in the model. Additionally, the goodness of fit model was assessed using the Hosmer-Lemeshow test, which compared the expected outcomes with the observed outcomes. The resulting p-value more than 0.05 indicate that the model adequately fit the data.

## Ethics approval

This study received ethical approval from two committees: the Medical Research and Ethics Committee (MREC) of the Ministry of Health Malaysia, with serial number NMRR ID-21-01983-7FT (IIR) (S3 Fig), and the Research Ethics Committee (REC) of Universiti Teknologi MARA Malaysia, with reference number REC/12/2021 (FB/67) (S4 Fig). The investigators ensured the anonymity and confidentiality of all participants throughout the study, following the principles outlined in the Declaration of Helsinki.

## Result

A total of 400 participants diagnosed with HIV from 2019 onward participated in this study. The prevalence of HIV late presenters was 60.0% (n = 240, 95% CI: 55.0–65.0). Table 1 shows the results of the sociodemographic profile and clinical characteristics of the study participants. The mean age of the participants was 30.29 (SD ±7.77) years old. The majority of the participants were male (96.5%), Malaysian (99.3%), and Malay (76.5%). Additionally, a significant proportion of the participants were single (88.3%), had attained a tertiary education level (72.5%), worked in the private sector (59.8%) and belonged to the low-income group (82.0%). Regarding the mode of transmission, most participants identified sexual transmission as the primary route. The majority of the sample consisted of MSM, accounting for 68.5%, followed by heterosexual individuals (12.5%), bisexual individuals (7.5%), PWID (1.3%), cases related to blood transfusion (0.8%) and unknown cases (9.5%). The mean score for HIV-related stigma among the participants was 107.25 (SD ±23.73) out of 160 (maximum score) and the mean score for HIV knowledge was 10.49 (SD ±3.80) out of 18 (maximum score).

In Table 2, the mean score of HIV-related stigma among HIV late presenters was 116.54 (±23.73), which was significantly higher compared to non-late presenters who scored 93.31 (±21.36), with a p-value of <0.001. Additionally, the mean scores of the four subscales of HIV stigma among late presenters were significantly higher than those among non-late presenters. The mean score for personalized stigma was 46.46 (±11.93), the mean score for disclosure stigma was 30.93 (±5.02), the mean score for negative self-image stigma was 35.28 (±7.26), and the mean score for public attitudes stigma was 55.59 (±11.91). Furthermore, the level of knowledge about HIV among late presenters was lower (mean score 9.00 ±3.32) than that among non-late presenters (mean score 12.71 ±3.3), with a p-value of <0.001.

**Table 1. Sociodemographic profile and clinical characteristics of study participants (N = 400).**

| Demographic characteristics | Overall, n (%) | HIV Late Presenters, n (%) | HIV Non-Late Presenters, n (%) |
|---|---|---|---|
| | 400 (100.0) | 240 (60.0) | 160 (40.0) |
| Mean Age (±SD) | 30.29 (±7.77) | 30.85 (±8.19) | 29.44 (±7.04) |
| Gender | | | |
| Male | 386 (96.5) | 232 (96.7) | 154 (96.3) |
| Female | 14 (3.5) | 8 (3.3) | 6 (3.8) |
| Nationality | | | |
| Malaysian | 397 (99.3) | 239 (99.6) | 158 (98.8) |
| Non-Malaysian | 3 (0.8) | 1 (0.4) | 2 (1.3) |
| Ethnicity | | | |
| Malay | 306 (76.5) | 188 (78.3) | 118 (73.8) |
| Chinese | 57 (14.2) | 35 (14.6) | 22 (13.8) |
| Indian | 12 (3.0) | 6 (2.5) | 6 (3.8) |
| Bumiputra Sarawak/Sabah | 25 (6.3) | 11 (4.6) | 14 (8.8) |
| Marital status | | | |
| Married | 38 (9.5) | 17 (7.1) | 21 (13.1) |
| Single | 353 (88.3) | 216 (90.0) | 137 (85.6) |
| Widowed/Divorced | 9 (2.3) | 7 (2.9) | 2 (1.3) |
| Education level | | | |
| Primary | 6 (1.5) | 2 (0.8) | 4 (2.5) |
| Secondary | 104 (26.0) | 58 (24.2) | 46 (28.7) |
| Tertiary | 290 (72.5) | 180 (75.0) | 110 (68.8) |
| Occupation | | | |
| Govt Sector | 37 (9.3) | 23 (9.6) | 14 (8.8) |
| Private Sector | 239 (59.8) | 143 (59.6) | 96 (60.0) |
| Self-employed | 64 (16.0) | 42 (17.5) | 22 (13.8) |
| Unemployed | 60(15.0) | 32 (13.3) | 28 (17.5) |
| Monthly income | | | |
| Low income | 328 (82.0) | 191 (79.6) | 137 (85.6) |
| Middle income | 65 (16.3) | 44 (18.3) | 21 (13.1) |
| High income | 7 (1.8) | 5 (2.1) | 2 (1.3) |
| Mode of transmission | | | |
| Heterosexual | 50 (12.5) | 29 (12.1) | 21 (13.1) |
| MSM | 274 (68.5) | 163 (67.9) | 111 (69.4) |
| Bisexual | 30 (7.5) | 19 (7.9) | 11 (6.9) |
| PWID | 5 (1.3) | 3 (1.3) | 2 (1.3) |
| Blood transfusion | 3 (0.8) | 1 (0.4) | 2 (1.3) |
| Unknown | 38 (9.5) | 25 (10.4) | 13 (8.1) |

MSM = men who have sex with men

PWID = people who inject drugs

The individual item statistics for the Malay version of the HIV-KQ-18 are presented in Table 3. Among the items, three recorded the lowest mean scores, ranging from 0.43 to 0.45. These items are: "There is a vaccine that can stop adults from getting HIV," "A person will NOT get HIV if she or he is taking antibiotics," and "A person can get HIV from oral sex." On the other hand, two items achieved the highest mean scores, ranging from 0.90 to 0.93. These

**Table 2. Comparison of the mean score of HIV related stigma and its subscales and the mean score of HIV knowledge between HIV late presenters and non-late presenters (N = 400).**

| | Overall | HIV late presenters | HIV non-late presenters | Mean difference (95% CI) | p-value |
|---|---|---|---|---|---|
| Mean HIV related stigma (±SD) | 107.25 (23.73) | 116.54 (20.48) | 93.31 (21.36) | 23.23 (19.04, 27.41) | <0.001 |
| Mean personalized stigma (±SD) | 44.20 (11.93) | 46.45 (11.76) | 40.82 (11.39) | 5.64 (3.31, 7.97) | <0.001 |
| Mean disclosure stigma (±SD) | 30.01 (5.56) | 30.93 (5.02) | 26.83 (6.03) | 2.29 (1.20, 3.39) | <0.001 |
| Mean negative self-image stigma (±SD) | 34.06 (7.54) | 35.28 (7.26) | 32.23 (7.60) | 3.05 (1.57, 4.53) | <0.001 |
| Mean public attitudes stigma (±SD) | 53.49 (12.40) | 55.59 (11.91) | 50.34 (12.49) | 5.25 (2.81, 7.69) | <0.001 |
| Mean HIV knowledge (±SD) | 10.49 (3.80) | 9.00 (3.32) | 12.71 (3.37) | -3.71 (-4.379, -3.038) | <0.001 |

*Independent t-test

**Table 3. Frequency of correct responses and mean score for each HIV-KQ-18 item among HIV late presenters (N = 240).**

| No | HIV KQ-18 Item | Mean item score (±SD) | Frequency (%) |
|---|---|---|---|
| Q1 | Coughing and sneezing DO NOT spread HIV. (T) | 0.85 (0.36) | 201 (83.7) |
| Q2 | A person can get HIV by sharing a glass of water with someone who has HIV. (F) | 0.90 (0.30) | 214 (89.2) |
| Q3 | Pulling out the penis before a man climaxes/cums keep a woman from getting HIV during sex. (F) | 0.54 (0.50) | 128 (53.3) |
| Q4 | A woman can get HIV if she has anal sex with a man. (T) | 0.66 (0.48) | 156 (65.0) |
| Q5 | Showering, or washing one's genitals/private parts, after sex keeps a person from getting HIV. (F) | 0.59 (0.50) | 139 (57.9) |
| Q6 | All pregnant women infected with HIV will have babies born with AIDS. (F) | 0.57 (0.50) | 134 (55.8) |
| Q7 | People who have been infected with HIV quickly show serious signs of being infected. (F) | 0.58 (0.50) | 137 (57.1) |
| Q8 | There is a vaccine that can stop adults from getting HIV. (F) | 0.43 (0.50) | 103 (42.9) |
| Q9 | People are likely to get HIV by deep kissing, putting their tongue in their partner's mouth, if their partner has HIV. (F) | 0.63 (0.48) | 150 (62.5) |
| Q10 | A woman cannot get HIV if she has sex during her period. (F) | 0.52 (0.50) | 124 (51.7) |
| Q11 | There is a female condom that can help decrease a woman's chance of getting HIV. (T) | 0.54 (0.50) | 128 (53.3) |
| Q12 | Using condom during sex prevents a person from getting HIV. *(T) | 0.93 (0.26) | 220 (91.7) |
| Q13 | A person will NOT get HIV if she or he is taking antibiotics. (F) | 0.44 (0.50) | 104 (43.3) |
| Q14 | Having sex with more than one partner can increase a person's chance of being infected with HIV. (T) | 0.89 (0.31) | 210 (87.5) |
| Q15 | Taking a test for HIV one week after having sex will tell a person if she or he has HIV. (F) | 0.50 (0.50) | 118 (49.2) |
| Q16 | A person can get HIV by sitting in a hot tub or a swimming pool with a person who has HIV. (F) | 0.81 (0.39) | 193 (80.4) |
| Q17 | A person can get HIV from oral sex. (T) | 0.45 (0.50) | 107 (44.6) |
| Q18 | Using Vaseline or baby oil with condoms lowers the chance of getting HIV. (F) | 0.47 (0.50) | 112 (46.7) |

Correct answers appear in parentheses (T = true; F = false).

*Added item for Malay version of HIV-KQ-18

items are: "A person can get HIV by sharing a glass of water with someone who has HIV" and "Using condom during sex prevents a person from getting HIV."

Table 4 presents the factors associated with HIV late presenters as determined through simple logistic regression and multiple logistic regression analyses. After adjusting for confounding risk factors, a high level of HIV-related stigma emerged as a risk factor for HIV late presenters (aOR = 1.049, 95% CI: 1.034–1.063, p-value <0.001). PLHIV with a high level of HIV knowledge exhibited 29.1% lower odds of presenting late for HIV than those with lower HIV knowledge (aOR = 0.709, 95% CI: 0.646–0.778, p-value <0.001). Patients with a tertiary education background were 16 times more likely to be diagnosed late for HIV in comparison to those with a primary education background (aOR = 15.962, 95% CI: 1.898–134.235, p-value = 0.011). Additionally, being single was identified as a risk factor for late presentation, with a 3.5-fold higher chance of being a late presenter compared to being married (aOR = 3.582, 95% CI: 1.393–9.208, p-value = 0.008).

## Discussion

Malaysia is committed to the UNAIDS global vision of ending AIDS as a public health threat by 2030. However, delays in HIV diagnosis can have a cascading impact on achieving the global target of 90-90-90 in the testing-treatment-viral suppression cascade. HIV late presenters may experience delays in initiating antiretroviral therapy (ART), which not only prolongs the period of potential viral transmission but also poses a greater challenge in achieving viral suppression [29, 30]. The early initiation of ART has shown a significant reduction in HIV transmission and clinical events, highlighting the benefits from both individual and public health perspectives [31].

The prevalence of HIV late presenters serves as a critical indicator for evaluating the effectiveness of testing and preventive initiatives. In our study, we found a considerably high prevalence of HIV late presenters at 60.0% (95% CI: 55.0–65.0). However, this prevalence is lower than that in some Asian countries, where it ranges from 70.2% to 72% [32–34]. The variation in prevalence could be attributed to differences in the sampling site among studies. In our study, we conducted sampling in public primary care clinic settings, whereas other studies utilized tertiary hospitals or national HIV surveillance systems. These variations in sampling sites and methodologies may account for the differences in reported prevalence rates.

These findings highlight the urgent need to address late HIV diagnosis and emphasize the importance of improved testing and awareness campaigns. Notably, our local national database from the Country Progress Report of Global AIDS Monitoring 2022 shows a similar pattern, with 68% of new patients being diagnosed at a late stage in 2021 [4]. On the other hand, resource-rich countries exhibit a relatively lower prevalence of late presentation, ranging from 49.8% to 50.4% [35, 36].

The mean age of HIV late presenters in this study was slightly older than that of non-late presenters. This finding confirms the results of numerous other studies, which consistently demonstrate that older age is associated with significantly higher odds of late presentation. In fact, the risk of late presentation is approximately twofold higher in older individuals [33, 34]. The significant proportion of participants in this study belonging to the low-income group (82%) highlights the financial burdens they face. This finding is valuable because it enables policymakers to develop policies that are not financially burdensome for this population, in line with the WHO agenda to achieve Universal Health Coverage (UHC). By ensuring equitable access to healthcare resources, policymakers can implement effective interventions [37]. However, the literature on the association between the low-income group and late presenters is limited.

**Table 4. Factors associated with HIV late presenters from multiple logistic regression analysis.** (Method Backward LR).

| Variable | Univariate analysis | | Multivariate analysis | |
|---|---|---|---|---|
| | p-value | Crude OR (95% CI) | p-value | Adjusted OR (95% CI) |
| HIV stigma | <0.001 | 1.054 (1.041, 1.067) | <0.001 | 1.049 (1.034, 1.063) |
| HIV knowledge | <0.001 | 0.708 (0.654, 0.767) | <0.001 | 0.709 (0.646, 0.778) |
| Age | 0.077 | 1.025 (0.997, 1.053) | - | - |
| Gender | | | | |
| Male | ref. | 1 | - | - |
| Female | 0.824 | 0.885 (0.301, 2.601) | | |
| Nationality | | | | |
| Malaysian | ref. | 1 | - | - |
| Non-Malaysian | 0.368 | 0.331 (0.030, 3.676) | | |
| Ethnicity | | | | |
| Malay | ref. | 1 | - | - |
| Chinese | 0.999 | 0.999 (0.599, 1.785) | | |
| Indian | 0.429 | 0.628 (0.198, 1.992) | | |
| Bumiputra Sabah/Sarawak | 0.092 | 0.493 (0.217, 1.123) | | |
| Marital status | | | | |
| Married | ref. | 1 | ref. | 1 |
| Single | 0.053 | 1.948 (0.992, 3.823) | 0.008 | 3.582 (1.393, 9.208) |
| Widowed/Divorced | 0.091 | 4.324 (0.793, 23.586) | 0.233 | 5.092 (0.351, 73.775) |
| Education level | | | | |
| Primary | ref. | 1 | ref. | 1 |
| Secondary | 0.298 | 2.522(0.442, 14.381) | 0.042 | 9.226(1.086, 78.389) |
| Tertiary | 0.175 | 3.273(0.590, 18.165) | 0.011 | 15.962(1.898, 134.235) |
| Occupation | | | | |
| Govt Sector | ref. | 1 | - | - |
| Private Sector | 0.788 | 0.907 (0.444, 1.850) | | |
| Self-employed | 0.726 | 1.162 (0.501, 2.695) | | |
| Unemployed | 0.395 | 0.696 (0.302, 1.605) | | |
| Monthly income | | | | |
| Low income | ref. | 1 | - | - |
| Middle income | 0.157 | 1.503 (0.855, 2.642) | | |
| High income | 0.489 | 1.793 (0.343, 9.379) | | |
| Mode of transmission | | | | |
| Heterosexual | ref. | 1 | - | - |
| MSM | 0.844 | 1.063 (0.577, 1.959) | | |
| Bisexual | 0.638 | 1.251 (0.493, 3.173) | | |
| PWID | 0.931 | 1.086 (0.167, 7.085) | | |
| Blood transfusion | 0.419 | 0.362 (0.031, 4.260) | | |
| Unknown | 0.458 | 1.393 (0.581, 3.339) | | |

MSM = men who have sex with men

PWID = people who inject drugs

*All significant independent variables in the multiple logistic regression were checked for multicollinearity using VIF. None of the variables showed multicollinearity (VIF less than 10).

**The model reasonable fits well (Hosmer-Lemeshow P value of 0.407). Model assumptions are met. There are no interaction problem.

***Variables with a p-value <0.25, including age, ethnicity, marital status, education level, monthly income, level of HIV-related stigma, and level of HIV knowledge were entered to MLogR analysis (Backward LR).

In recent years, the epidemic landscape of HIV transmission has shifted from people who inject drugs (PWID) to HIV infection through sexual transmission. The ratio of transmission attributed to PWID compared to sexual transmission has decreased from 3.95 in 2000 to 0.04 in 2021 [4]. Our study revealed that sexual transmission accounted for approximately 88.5% of the overall mode of transmission, with MSM exhibiting the highest frequency at 68.5%. This finding underscores the critical role of targeting MSM as a key population in combating the AIDS epidemic. MSM face unique challenges in HIV prevention and treatment due to various factors, including social stigma, discrimination, and limited access to healthcare services [38]. It is crucial to develop targeted interventions and support systems that address the specific needs of this key population [39]. Additionally, attention should be given to heterosexual transmission, as numerous studies have found a significant association between heterosexual transmission and late presenters [34, 36].

The mean score of HIV-related stigma among all participants is consistent with the results of a previous local study conducted by Abdullah, A. (2020). That study reported a mean score of 104.7 (±19.5) [40]. Our study identifies HIV late presenters as having a higher likelihood of experiencing high levels of HIV-related stigma. This outcome corroborates findings from other studies, which showed a two to three times higher likelihood of being late presenters [10, 12, 41]. The intensified levels of stigma and discrimination experienced by HIV late presenters can create substantial barriers to seeking timely diagnosis and care, leading affected individuals to exhibit reluctance in engaging with healthcare services, disclosing their HIV status, or accessing the necessary support systems [12].

The mean HIV knowledge score among late presenters was significantly lower than that of non-late presenters. This significant difference in mean scores indicates a notable disparity in knowledge, suggesting that late presenters may have limited awareness about HIV transmission. PLHIV with a high level of HIV knowledge demonstrated a 29.1% reduction in the likelihood of presenting late for HIV compared to those with lower HIV knowledge, supporting previous study findings. Beyene et al. reported a 74% lower likelihood of late diagnosis for PLHIV with high knowledge. However, their study used low, medium, and high knowledge categories, while ours employed mean scores for comparison, possibly explaining the difference in odds ratio [42]. These gaps in knowledge can contribute to delays in seeking testing or care, ultimately resulting in late HIV presentation [10].

A higher educational background is commonly regarded as a protective factor against late HIV presentation [6, 43, 44]. However, our study yielded unexpected findings. We discovered that PLHIV who possessed a tertiary education background were 16 times more likely to be diagnosed late for HIV than those with a primary education background. While we could not find existing research to directly support this association, one possible explanation for this unexpected result is our hypothesis that highly educated PLHIV may have concerns that their HIV status could adversely affect their relationships, career prospects, or livelihoods, thereby contributing to delayed diagnosis. Stigma associated with HIV could potentially hinder their willingness to seek testing or care due to the fear of being judged or facing discrimination [10, 12].

Being single was found to be a risk factor for late HIV presentation, with a 3.5-fold higher likelihood of being a late presenter compared to those who were married. One possible explanation for this correlation is that in Malaysia, there is no requirement for single individuals to undergo HIV screening, unlike married individuals specifically Muslim couples who are obligated to undergo premarital testing [45]. Unless single individuals proactively come forward for testing due to high-risk behavior, they remain undiagnosed until presenting at a late stage. It is worth noting that apart from premarital testing, antenatal mothers are routinely screened for HIV during their booking visit [46].

## Limitations of the study

The limitation of the study could be sampling bias. The research findings may not be generalizable to the entire population, as the study sample might not adequately represent the diversity of the larger population. Another limitation could be reliance on self-reported data, particularly their clinical characteristics which introduces the potential of recall bias and social desirability bias. Participants might have provided responses that they perceived as socially acceptable rather than reflecting their true self-perception, which could impact the reliability and validity of the findings. The study might have focused on specific risk factors related to HIV late presentation, such as stigma and HIV knowledge, while not considering other potentially influential factors. Factors such as access to health care and mental health status could play a role in HIV late presentation but were not examined in this study.

## Implication of clinical practice and future research

Our research findings suggest several key implications. First, healthcare providers should prioritize implementing stigma reduction interventions to create a safe and supportive environment for individuals seeking HIV testing and treatment. This can be achieved by training healthcare providers to confront and overcome stigma, providing counselling services, and fostering open and non-judgemental communication.

Second, HIV education and awareness should be emphasized to improve knowledge on HIV. Healthcare providers play a vital role in disseminating accurate and up-to-date information about HIV transmission, prevention, and available healthcare services. This can be achieved through public educational programs, community outreach initiatives, and offer screening protocols among key population groups.

Finally, targeted interventions for key populations, such as MSM, should be developed. These interventions should address their specific needs and challenges related to HIV testing and late presentation. Tailored educational campaigns, peer support programs, and strategies to improve access to healthcare services can contribute to early diagnosis and timely access to care.

Future research should aim to explore the cultural and contextual factors that influence HIV-related stigma and knowledge, as these factors can vary across different populations and settings. It is important to note that Malaysia, being a multicultural country with diverse ethnicities, may have unique factors that differ from those of Western populations. Understanding these nuances will help in developing culturally sensitive interventions to address specific challenges faced by diverse communities.

Additionally, evaluating the effectiveness of stigma reduction interventions, HIV education programs, and targeted interventions for key populations is essential. Assessing the impact of these interventions on reducing late HIV presentation will help in refining and implementing evidence-based practices in clinical settings.

## Conclusion

The presence of stigma surrounding HIV/AIDS can lead to fear, discrimination, and reluctance to seek testing and treatment, resulting in delayed diagnosis. Inadequate knowledge about HIV transmission, prevention, and available healthcare services further contributes to late presentation. To address these challenges, it is crucial to implement targeted efforts such as community outreach initiatives aimed at reducing stigmaand improving HIV education. By promoting acceptance, dispelling misconceptions, and enhancing access to accurate information, we can encourage early HIV testing and timely access to care, leading to improved health outcomes for PLHIV.

## Supporting information

**S1 Fig. Permission to use Berger HIV stigma scale questionnaire.**
(DOCX)

**S2 Fig. Permission to use brief HIV-KQ-18 questionnaire.**
(DOCX)

**S3 Fig. Ethical approval the Medical Research and Ethics Committee (MREC) of the Ministry of Health Malaysia.**
(DOCX)

**S4 Fig. Research Ethics Committee (REC) of Universiti Teknologi MARA Malaysia approval.**
(DOCX)

## Acknowledgments

The authors would like to express their gratitude to the grant provider: UiTM Selangor Branch Research Grant Fund (DUCS) 4.0: Grant code: 600-UiTMSEL (PI. 5/4) (024/2022), and to the developers for granting permission to utilize their questionnaires in this study. We extend our heartfelt thanks to the Family Medicine Specialists and clinical staff in their respective clinics for their valuable assistance and cooperation.

## Author Contributions

**Conceptualization:** Mohd Zulfikry Bin Ahmad.

**Data curation:** Mohd Zulfikry Bin Ahmad.

**Formal analysis:** Mohd Zulfikry Bin Ahmad, Mariam Mohamad.

**Funding acquisition:** Mazapuspavina Md. Yasin.

**Investigation:** Mohd Zulfikry Bin Ahmad.

**Methodology:** Mohd Zulfikry Bin Ahmad.

**Project administration:** Mohd Zulfikry Bin Ahmad, Mazapuspavina Md. Yasin.

**Supervision:** Mazapuspavina Md. Yasin, Nafiza Mat Nasir, Mariam Mohamad.

**Validation:** Mazapuspavina Md. Yasin.

**Writing – original draft:** Mohd Zulfikry Bin Ahmad, Mazapuspavina Md. Yasin.

**Writing – review & editing:** Mohd Zulfikry Bin Ahmad, Mazapuspavina Md. Yasin, Nafiza Mat Nasir, Mariam Mohamad.

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
