## [Decision Letter · Decision Letter 0]

13 Sep 2023

PONE-D-23-24306The Association Between HIV-Related Stigma, HIV Knowledge and HIV Late Presenters Among People Living with HIV (PLHIV) Attending Public Primary Care Clinic Settings in SelangorPLOS ONE

Dear Dr. Md-Yasin,

Thank you for submitting your manuscript to PLOS ONE. After careful consideration, we feel that it has merit but does not fully meet PLOS ONE’s publication criteria as it currently stands. Therefore, we invite you to submit a revised version of the manuscript that addresses the points raised during the review process.

We look forward to receiving your revised manuscript.

Kind regards,

Nelsensius Klau Fauk, S.Fil., M., MHID, MSc, PhD

Academic Editor

PLOS ONE

Journal Requirements:

   "Funded studies

Authors received award: Mazapuspavina Md Yasin, Nafiza Mat Nasir, Mariam Mohamad

Grant number: 600-UiTMSEL (PI. 5/4) (024/2022)

Geran Penyelidikan Dana UiTM Cawangan Selangor (DUCS) Malaysia

URL: https://orchid.uitm.edu.my/irmis/index.php?val=loggedout

NO role"

Additional Editor Comments:

Additional comments to the authors

Abstract

• The definition of late presenter should be placed right at the beginning or in the introduction section.

• The authors recommended “Effective intervention programs targeting early HIV diagnosis, improved HIV care, and treatment are necessary to address this issue” but didn’t think of factors that prevent early HIV diagnosis and treatment as they reported in the findings. Think about this and address it.

INTRODUCTION

• Please use the current 2022 UNAIDS data that are available online.

• Similarly, use current HIV data about Malaysia, 2021 data are nearly 2 years ago.

• It is a very long introduction which contains some unnecessary information. I would suggest deleting lines 54-67. This paragraph talks about the impact of HIV or late presentation for HIV care, while the focus of this study is on investigating factors associated with late presentation for HIV care.

• My biggest concern about the introduction section is that the synthesis of the existing findings from previous studies on factors associated with or influencing the presentation or access of PLHIV to HIV care or treatment is poorly done (see lines 68-95). There have been massive studies on this aspect globally and in Asia, which the authors need to consult and provide a good synthesis. Some resources are recommended below.

• The synthesis will help the authors to finally formulate the knowledge gap they wanted to bridge through this study or to state the novelty of their study and why it is important. This will become your contribution to knowledge.

• The authors can consult and use the following resources, which report on HIV stigma, HIV knowledge and various factors that influence access to HIV testing, care and treatment among PLHIV:

o HIV Stigma and Moral Judgement: Qualitative Exploration of the Experiences of HIV Stigma and Discrimination among Married Men Living with HIV in Yogyakarta. Int. J. Environ. Res. Public Health 2020, 17, 636. https://doi.org/10.3390/ijerph17020636
https://www.mdpi.com/1660-4601/17/2/636

o Stigma and Discrimination towards People Living with HIV in the Context of Families, Communities, and Healthcare Settings: A Qualitative Study in Indonesia. Int. J. Environ. Res. Public Health 2021, 18, 5424. https://doi.org/10.3390/ijerph18105424
https://www.mdpi.com/1660-4601/18/10/5424

o HIV stigma and discrimination: Perspectives and personal experiences of healthcare providers in Yogyakarta and Belu. Frontiers in Medicine 8, 625787. https://www.frontiersin.org/articles/10.3389/fmed.2021.625787/full

o Barriers to HIV testing among male clients of female sex workers in Indonesia. Int J Equity Health 17, 68 (2018). https://doi.org/10.1186/s12939-018-0782-4
https://equityhealthj.biomedcentral.com/articles/10.1186/s12939-018-0782-4#citeas

o The Intention of Men Who Have Sex With Men to Participate in Voluntary Counseling and HIV Testing and Access Free Condoms in Indonesia. American Journal of Men’s Health. 2018;12(5):1175-1184. doi:10.1177/1557988318779737 https://journals.sagepub.com/doi/full/10.1177/1557988318779737

o Barriers to Accessing HIV Care Services in Host Low and Middle Income Countries: Views and Experiences of Indonesian Male Ex-Migrant Workers Living with HIV. Int. J. Environ. Res. Public Health 2022, 19, 14377. https://doi.org/10.3390/ijerph192114377
https://www.mdpi.com/1660-4601/19/21/14377

o Traditional Human Immunodeficiency Virus treatment and family and social influence as barriers to accessing HIV care services in Belu, Indonesia. PLoS ONE 17(7): e0264462. https://doi.org/10.1371/journal.pone.0264462
https://journals.plos.org/plosone/article?id=10.1371/journal.pone.0264462

DISCUSSION

The authors stated: “To the best of our knowledge, this is the first study in Malaysia to determine the prevalence and explore the risk factors for HIV late presenters”. I do encourage the authors to consult the existing literature on this topic in Malaysia and globally and discuss your findings in light of previous findings, theories, etc. Please consult the following resources:

• Delayed HIV testing and treatment seeking, and associated support needs among people living with HIV in Malaysia: a qualitative study. https://www.proquest.com/openview/bf841f3f539304a03b795ca784414a89/1?pq-origsite=gscholar&cbl=2037399

• Factors Affecting Voluntary HIV Testing Among General Adult Population: A Cross-Sectional Study in Sarawak, Malaysia. https://www.ncbi.nlm.nih.gov/pmc/articles/PMC7428414/

• Late Presentation into Care of HIV Disease and Its Associated Factors in Asia: Results of TAHOD. https://www.ncbi.nlm.nih.gov/pmc/articles/PMC4779961/

• HIV Testing and awareness of HIV status among people who inject drugs in greater Kuala Lumpur, Malaysia. https://www.tandfonline.com/doi/abs/10.1080/09540121.2017.1363852

• Drivers of HIV self-testing among female sex workers: Findings from a multi-state study in Malaysia. https://www.frontiersin.org/articles/10.3389/fmed.2023.1022746/full

Additional resources:

• HIV/AIDS late presentation and its associated factors in China from 2010 to 2020: a systematic review and meta-analysis. AIDS Res Ther 18, 96 (2021). https://doi.org/10.1186/s12981-021-00415-2
https://aidsrestherapy.biomedcentral.com/articles/10.1186/s12981-021-00415-2#citeas

• Late presentation of HIV positive adults and its predictors to HIV/AIDS care in Ethiopia: a systematic review and meta-analysis. BMC Infect Dis 19, 534 (2019). https://doi.org/10.1186/s12879-019-4156-3
https://bmcinfectdis.biomedcentral.com/articles/10.1186/s12879-019-4156-3#citeas

Reviewers' comments:

Reviewer's Responses to Questions

**Comments to the Author**

1. Is the manuscript technically sound, and do the data support the conclusions?

Reviewer #1: Yes

Reviewer #2: Yes

2. Has the statistical analysis been performed appropriately and rigorously? 

Reviewer #1: I Don't Know

Reviewer #2: Yes

3. Have the authors made all data underlying the findings in their manuscript fully available?

Reviewer #1: Yes

Reviewer #2: Yes

4. Is the manuscript presented in an intelligible fashion and written in standard English?

Reviewer #1: Yes

Reviewer #2: Yes

5. Review Comments to the Author

Reviewer #1: I think it's a very important study, a well-reviewed and well-designed study.

I would like to point out a few things about the statistical analysis.

1. About multivariate modeling. The author states that the modeling of multivariate analysis is based on the diagnosis of multicollinearity and a model fit examination (Hosmer-Lemeshow test). However, the differences between the variables in the univariate model and multivariate model in Table 4 are not clearly explained. Based on the results of the logistic regression analysis, my guess is that non-significant variables were not included in the multivariate analysis model. Am I wrong?

Since multivariate adjustment is based on modeling and not on statistical probability and significance, I recommend avoiding the above method.

Also, variables that are likely to have multicollinearity should be carefully considered and adjusted, rather than just being deleted.

First, the authors should clearly state the results of the diagnosis of collinearity. At least VIFs greater than or equal to 10 should be removed from the multivariate model. Therefore, authors need to be clear about which variables result in high VIF.

After that, please consider the following. For example, there seems to be a high correlation between gender and marital status and MSM. There also seems to be a high correlation between Nationality and Ethnicity.

The same goes for income, occupation, and education.

These relationships can be easily inferred theoretically.

As mentioned above, I strongly recommend that you select the collinearity diagnostic results and the variables that should be adjusted content-wise and theoretically.

This moderating variable is very important in cross-sectional research designs, so please consider it carefully in the Method section.

In multivariate analysis, we strongly recommend adjusting the confounding variables using the forced entry method.

Considering the model of this study, I thought it would be better to present the results using hierarchical multiple logistic regression analysis.

2. The year of diagnosis or the number of years since diagnosis should be used as an adjustment variable.

This study is a cross-sectional study, and we do not know whether current knowledge and stigma arose after or before diagnosis.

Since you should at least know the year of diagnosis, we strongly recommend adjusting the year of diagnosis (or years since diagnosis).

3. Please indicate AIC, Hosmer-Lemeshow test, model chi-square value, and chi-square test results in the results section as model fit indicators for multivariate analysis results.

Reviewer #2: Thank you for sending this intriguing paper about HIV stigma and knowledge in relation to late presentation of PLHIV attending public primary care clinics in Selangor.

Late treatment or intervention for a condition results in a dismal prognosis. Consequently, this research is essential for enhancing case management. The fact that this study utilized a cross-sectional design with a large sample size (400 participants) demonstrated the study's power.

Here are my comments on this paper:

Abstract:

- Results: the authors provided the prevalence of late presenters (60%) with a confidence interval (CI) ranging from 55 to 65. As this is a prevalence, I cannot figure out how the authors derive the values of CI. Is it an aggregated value of the prevalence from many study settings so that they have the CI values? (the same thing later in the result section, while in the table 1 it was only the prevalence (n and %))

- Results: please re-check the results from the “tertiary education background” as I am not quite sure with the values provided there (aOR or the CI). Need to check the raw results from the software.

Introduction:

- First sentence (line 31-32) need a reference

- Stigma is not always imposed by outside parties. Internal stigma can also take place. The authors must provide information regarding this and the area in which they are concentrating their research.

Methods:

- Line 116-117: “Multiple sample sizes were calculated based on different prevalence…” Please elaborate more this sentence, what did they mean by “different prevalence”? (from previously published studies?)

- Why “2019 onward”? Is it because of the country progress report 2019 or any other reason?

- Its good that they have provided the detailed information of the tools, including the evaluation of the performance for both instruments

Results:

- Table 1: check the values of the percentages of some variables (not a 100% in total), i.e. males and females in the non-late presenters

- Table 4: re-check the values of ORs and Cis (some of the values are weird)

- Until the end of the results I could not find the CI values for the prevalence as they mentioned in the abstract, results (narrative), and discussion

Discussion:

- - It is preferable to use fewer numbers in the discussion (numbers have been presented in the results).

6. PLOS authors have the option to publish the peer review history of their article (what does this mean?). If published, this will include your full peer review and any attached files.

Reviewer #1: No

Reviewer #2: No

---

## [Author Response · Author response to Decision Letter 0]

13 Mar 2024

Dear Editorial Board,

We have addressed and corrected the manuscript based on the reviewers comments accordingly. 

All relevant revised/updated version has been uploaded. 

Thank you.

---

## [Decision Letter · Decision Letter 1]

9 Apr 2024

PONE-D-23-24306R1The association between HIV-related stigma, HIV knowledge and HIV late presenters among people living with HIV (PLHIV) attending public primary care clinic settings in SelangorPLOS ONE

Dear Dr. Md-Yasin,

Thank you for submitting your manuscript to PLOS ONE. After careful consideration, we feel that it has merit but does not fully meet PLOS ONE’s publication criteria as it currently stands. Therefore, we invite you to submit a revised version of the manuscript that addresses the points raised during the review process.

We look forward to receiving your revised manuscript.

Kind regards,

Nelsensius Klau Fauk, S.Fil., M., MHID, MSc, PhD

Academic Editor

PLOS ONE

Journal Requirements:

Additional Editor Comments:

The authors have sufficiently addressed the reviewers' comments in round 1. Please address a few further minor comments and resubmit.

Reviewers' comments:

Reviewer's Responses to Questions

**Comments to the Author**

1. If the authors have adequately addressed your comments raised in a previous round of review and you feel that this manuscript is now acceptable for publication, you may indicate that here to bypass the “Comments to the Author” section, enter your conflict of interest statement in the “Confidential to Editor” section, and submit your "Accept" recommendation.

Reviewer #1: (No Response)

2. Is the manuscript technically sound, and do the data support the conclusions?

Reviewer #1: Yes

3. Has the statistical analysis been performed appropriately and rigorously? 

Reviewer #1: Yes

4. Have the authors made all data underlying the findings in their manuscript fully available?

Reviewer #1: Yes

5. Is the manuscript presented in an intelligible fashion and written in standard English?

Reviewer #1: Yes

6. Review Comments to the Author

Reviewer #1: Thank you for giving me the opportunity to review.

I think this is very important and valuable research.

I would like to point out a few things.

1. Please be more specific and clear about your research objective, L121~123. "...including HIV-related stigma and knowledge", but isn't this study focusing on both of these as independent variables?

2. Please explain how missing values are handled during analysis, along with its validity.

3. L250, The author sets the VIF standard at 10. Please provide the basis for that.

4. About modeling in logistic regression analysis. It is theoretically insufficient to omit multivariate adjustment on the basis that the relationship between the adjustment variable and the dependent variable is not statistically significant in univariate analysis.

Since multicollinearity has already been taken into account, it is necessary to insert adjustment variables into the model and actively adjust the results, regardless of statistical significance, and present the results as the final model.

7. PLOS authors have the option to publish the peer review history of their article (what does this mean?). If published, this will include your full peer review and any attached files.

Reviewer #1: No

---

## [Author Response · Author response to Decision Letter 1]

23 Jun 2024

Dear Editorial Board

Thank you for the comments given. The comments have need addressed in the newly submitted corrected manuscript V3. Appreciate positive reply.

---

## [Editor Report · Decision Letter 2]

26 Jun 2024

The association between HIV-related stigma, HIV knowledge and HIV late presenters among people living with HIV (PLHIV) attending public primary care clinic settings in Selangor.

PONE-D-23-24306R2

Dear Dr. Mazapuspavina,

We’re pleased to inform you that your manuscript has been judged scientifically suitable for publication and will be formally accepted for publication once it meets all outstanding technical requirements.

Kind regards,

Nelsensius Klau Fauk, S.Fil., M., MHID, MSc, PhD

Academic Editor

PLOS ONE
---

## [Editor Report · Acceptance letter]

11 Jul 2024

PONE-D-23-24306R2 

PLOS ONE

Dear Dr. Md-Yasin, 

I'm pleased to inform you that your manuscript has been deemed suitable for publication in PLOS ONE. Congratulations! Your manuscript is now being handed over to our production team.

Kind regards, 

on behalf of

Dr. Nelsensius Klau Fauk 

Academic Editor

PLOS ONE